# The Novel Synthetic Antibiotic BDTL049 Based on a Dendritic System Induces Lipid Domain Formation while Escaping the Cell Envelope Stress Resistance Determinants

**DOI:** 10.3390/pharmaceutics15010297

**Published:** 2023-01-16

**Authors:** Philipp F. Popp, Tania Lozano-Cruz, Franziska Dürr, Addis Londaitsbehere, Johanna Hartig, Francisco Javier de la Mata, Rafael Gómez, Thorsten Mascher, Ainhoa Revilla-Guarinos

**Affiliations:** 1Department of General Microbiology, Institut Für Mikrobiologie, Technische Universität Dresden, 01217 Dresden, Germany; 2Department of Organic and Inorganic Chemistry, Research Institute in Chemistry “Andrés M. Del Río” (IQAR), University de Alcalá, 28805 Madrid, Spain; 3Ramón y Cajal Health Research Institute (IRYCIS), 28805 Madrid, Spain; 4Networking Research Center on Bioengineering, Biomaterials and Nanomedicine (CIBER-BBN), 28805 Madrid, Spain

**Keywords:** drug design, click chemistry, mode of action, cell envelope stress response, carbosilane dendritic system, *Bacillus subtilis*, antimicrobial resistance, fluorescent microscopy, membrane labeling, novel synthetic antibiotic

## Abstract

The threat of antimicrobial-resistant bacteria is ever increasing and over the past-decades development of novel therapeutic counter measurements have virtually come to a halt. This circumstance calls for interdisciplinary approaches to design, evaluate and validate the mode of action of novel antibacterial compounds. Hereby, carbosilane dendritic systems that exhibit antimicrobial properties have the potential to serve as synthetic and rationally designed molecules for therapeutic use. The bow-tie type topology of BDTL049 was recently investigated against the Gram-positive model organism *Bacillus subtilis*, revealing strong bactericidal properties. In this study, we follow up on open questions concerning the usability of BDTL049. For this, we synthesized a fluorescent-labeled version of BDTL049 that maintained all antimicrobial features to unravel the interaction of the compound and bacterial membrane. Subsequently, we highlight the bacterial sensitivity against BDTL049 by performing a mutational study of known resistance determinants. Finally, we address the cytotoxicity of the compound in human cells, unexpectedly revealing a high sensitivity of the eukaryotic cells upon BDTL049 exposure. The insights presented here further elaborate on the unique features of BDTL049 as a promising candidate as an antimicrobial agent while not precluding that further rounds of rational designing are needed to decrease cytotoxicity to ultimately pave the way for synthetic antibiotics toward clinical applicability.

## 1. Introduction

In the context of a post-global pandemic and a global antibiotic crisis, with the threatening increase of multidrug resistance bacteria, research on new antimicrobial agents is one of the priorities of the Global Action Plan on Antimicrobial Resistance developed by the World Health Organization [1]. 

Different strategies can be used for the development of new antimicrobials, including *(i)* identifying new natural compounds (such as antimicrobial peptides produced, for example, by lactic acid bacteria [2,3], or secondary metabolites produced by *Streptomyces* [4,5]) *(ii)* chemical enhancement of structure or delivery of existing antibiotics to boost their stability and potency [6,7], and *(iii)* exploit our current knowledge on natural antibiotics’ properties to rationally design novel antimicrobial agents [8,9]. Regarding the latter strategy, dendritic systems are a promising group of molecules for the development of novel synthetic antimicrobials [10]. 

Monodisperse dendrimers present a highly precise skeleton structure that is chemically synthesized from a central core and the subsequent addition of repeating units in a radial manner (increasing the molecule *generation* with each additional branching point) [11]. The *generation* of the system determines the final number of peripheral terminal groups. In the case of carbosilane dendrimers, the branched inner scaffold is constituted by carbon and silicon atoms which account for its lipophilic nature. The peripheral groups define the valency of the dendrimer, and its chemical nature determines the solubility and future application of these macromolecules. For example, the activity of cationic carbosilane dendritic systems with varied peripheral groups has been investigated for their use as antibacterials [12,13,14], antifungals [15], and antiamoebics [16,17].

We recently reported *de novo* rational design, the in vitro chemical synthesis, and the in vivo experimental testing of the new antimicrobial agent termed BDTL049 [18]. Based on a first-generation carbosilane scaffold, it is a bow-tie topology ammonium-terminated dendrimer with a hydrophobic inner skeleton but a positively charged periphery. BDTL049 has antimicrobial activity against Gram-positive and Gram-negative bacteria at concentration ranges comparable to clinically established natural antibiotics. A comprehensive analysis of the mode of action of BDTL049 using the Gram-positive model bacterium *Bacillus subtilis* showed that the new compound induces a strong cell envelope stress response while targeting primarily the cytoplasmic membrane where it causes a collapse of the membrane potential by pore-formation. However, several questions regarding the activity of BDTL049 remained elusive in our previous work, such as *(i)* does the compound remain on the cell surface, or does it penetrate into the cytoplasm affecting other cellular functions?; *(ii)* are the cell envelope response systems —which usually counteract the activity of naturally occurring antibiotics— able to provide resistance against the novel compound?; and last but not least, *(iii)* what is the cytotoxicity of the new compound against eukaryotic cells —which will determine its biocompatibility and future clinical applicability—? 

This new study aims at answering these questions. A green-labeled form of BDTL049, named BDTL100, was synthesized (Figure 1) and validated regarding retaining equal toxicity and mode of action as the parental compound (Figure 2). BDTL100 was used in fluorescent microscopy localization studies, which revealed its accumulation at the cytoplasmic membrane of the bacterial cells leading to lipid domain formation (Figure 3), and supported our previous observations pointing toward a heterogeneous affectation of the bacterial cell population. Interestingly, even though BDTL049 induces a strong cell envelope stress response in *B. subtilis* (as shown by previous transcriptomic analysis [18]), the sensitivity against BDTL049 of several mutant strains revealed that, except for a minor contribution of the Dlt system (Figure 4), the resistance determinants which mediate the cell envelope stress response to other antibiotics are unable to mount an effective resistance response against the novel compound leaving the cell rather defenseless. Finally, the biocompatibility of BDTL049 was investigated by determining its cytotoxicity in human cell lines. Our experimental results with the rationally designed novel antibiotic BDTL049 add valuable information for the knowledge-based strategy which seeks the development of increasingly potent antimicrobial molecules.

## 2. Materials and Methods

### 2.1. Chemical Synthesis of BDTL049 and BDTL100

The dendritic compounds used in this work (Figure 1) were synthesized following a similar (BDTL049) [18] or adapted (BDTL100) [19] protocol described previously. For BDTL100 (131.1 mg; 0.131 mmol), the thiol-ene reaction was carried out with two thiol derivatives. Firstly, cysteamine hydrochloride (11.9 mg; 0.104 mmol) was added and allowed to react for 30 min in the presence of DMPA (2.4 mg; 0.094 mol) before the addition of the 2-dimethyl-aminoethanethiol hydrochloride in the conditions employed in the literature. Then, the crude was neutralized in CH_2_Cl_2_/H_2_O(Na_2_CO_3_), and fluorescein was incorporated (13.4 mg; 0.034 mmol) in the primary amine using ethanol as solvents (16 h; room temperature). After filtering and evaporation, the labeled bow-tie dendrimer was dissolved in THF, and the amino groups were again protonated in the presence of hydrogen chloride (2M in Et_2_O; 1.834 mmol), dialyzed (MWCO: 100–500) and washed with ethanol.

### 2.2. Bacterial Strains and Growth Conditions

Table 1 lists the strains used in this study. *B. subtilis* cells were routinely grown in Luria-Bertani (LB-Medium (Luria/Miller), Carl Roth, Karlsruhe, Germany) medium at 37 °C with agitation. 1.5% (*w*/*v*) agar (Agar-Agar Kobe I, Carl Roth, Karlsruhe, Germany) was added to prepare the corresponding solid media. Strains were stored at −80 °C in their corresponding growth media containing 20% (*v*/*v*) glycerol (Carl Roth, Karlsruhe, Germany). Chloramphenicol (Sigma-Aldrich, Merck KGaA, Darmstadt, Germany) 5 µg mL^−1^, kanamycin (Carl Roth, Karlsruhe, Germany) 10 µg mL^−1^, tetracycline hydrochloride (Sigma-Aldrich, Merck KGaA, Darmstadt, Germany) 12.5 µg mL^−1^, erythromycin (Sigma-Aldrich, Merck KGaA, Darmstadt, Germany) 1 µg mL^−1^ and lincomycin (Alfa Aesar™ Lincomycin hydrochloride, Fisher Scientific, Kandel, Germany) 25 µg mL^−1^ were added to *B. subtilis* when required.

### 2.3. Creation luxABCDE Reporter Strains

Both promoters of the *pspA* gene and *liaIH* operon were PCR amplified from the *B. subtilis* genome and, via restriction digestion and ligation, inserted into the pBS3Clux reporter backbone [25]. Positive clones were verified by sequencing and subsequently integrated into the *B. subtilis* genome via homologous recombination. The primers used are listed in Table 2. 

### 2.4. Creation of Mutant Strains by Allelic Replacement Mutagenesis

The *dlt* operon from *B. subtilis* W168 contains five genes (*dltA-dltE*). Since the gene product of *dltE* is not involved in the D-alanilation of lipoteichoic acids (LTA) nor wall teichoic acids (WTA) [26], we created a *B. subtilis dlt* mutant eliminated in genes from *dltA* to *dltD*. The mutant strains eliminated in genes *dltABCD* and *mprF* were created by allelic replacement mutagenesis in *B. subtilis* W168 using long flanking homology (LFH)-PCR [27]. The genes were replaced by kanamycin (*kan*) and macrolide-lincosamide-streptogramine (*mls*) resistance cassettes, respectively. The vectors used as templates for the resistance cassettes are listed in Table 1. The procedure was performed as described previously [28]. Primer pairs used for amplification of the *kan* and *mls* cassettes, up- and down-fragments, and primers used to check the allelic replacement are listed in Table 2. 

### 2.5. Sensitivity and Promoter Induction Assays with Exponentially Growing Planktonic Cultures

All the experiments testing the sensitivity of exponentially growing cultures to BDTL049 and BDTL100 were performed in Mueller Hinton (MH) Broth for antibiotic-sensitivity testing [beef infusion 2 g L^−1^, casein peptone (acidic hydrolysate) 17.5 g L^−1^, corn starch 1.5 g L^−1^, and pH value 7.4 ± 0.2; Carl Roth, Karlsruhe, Germany]. The experiments testing the sensitivity of W168 cultures were performed with at least three biological replicas on different days, and the experiments with the mutant strains were performed with two biological clones and with two technical replicas on different days. The overnight cultures (3 mL) were prepared by picking single colonies from fresh plates, with antibiotic selection added when required. The day cultures (10 mL) were inoculated 1:200 with overnight cultures without antibiotic selection and incubated at 37 °C (220 rpm) until an OD_600_ of around 0.2 was reached. Then, the cell suspensions were diluted to an OD_600_ of 0.01, and they were distributed into a 96-well transparent plate (95 μL per well) and incubated at 37 °C (continuous middle shacking) in the Synergy™ NeoalphaB plate reader (BioTek^®^, Winooski, VT, USA). After one hour of incubation, 5 μL of the synthetic compounds (at 20 times the desired final concentrations) were added to the wells, with one well left untreated as a control. The incubation at 37 °C with continuous middle shaking was continued for a further 18h. OD_600_ was measured every 5 min to monitor the growth rate. 

For the whole-cell biosensors induction assays, the antibiotic selection was added to the overnight cultures, but the day cultures (10 mL) were inoculated 1:200 with the overnight cultures without antibiotic selection. The experiments with the reporter strains were performed in triplicate with two biological clones and with two technical replicas on different days. The same procedure as for testing the sensitivity of exponentially growing cultures was followed to determine the induction of the promoter-*luxABCDE* transcriptional fusions, but the cells were plated in black 96-wells plates (black, clear bottom; Greiner Bio-One, Frickenhausen, Germany) and besides OD_600_, luminesce between 300 to 700 nm was monitored every 5 min for at least 18 h. 

### 2.6. Determination of Minimal Inhibitory Concentration and Minimal Bactericidal Concentration 

All the sensitivity experiments for determination of the Minimal Inhibitory Concentration (MIC) and Minimal Bactericidal Concentration (MBC) were performed in Mueller-Hinton (MH) Broth (Carl Roth, Karlsruhe, Germany), as previously described [18]. The experiments were performed with at least three biological replicas on different days in the case of the wild-type strain and in biological duplicates (two clones) and technical triplicates (on different days) in the case of the mutant strains. Briefly, the overnight cultures (3 mL) of the strains under study were prepared, with antibiotic selection added when required, by picking single colonies from fresh plates. The day cultures were inoculated from freshly grown overnight cultures at an OD_600_ of 0.05 in fresh medium containing 2-fold serial dilutions of the compound under study but no antibiotics for selection. The day cultures were plated into 96-well microtiter plates (100 µL per well), and the plates were incubated at 37 °C with constant middle shacking in a Synergy^TM^ NEOALPHAB multi-mode microplate reader (BioTek^®^, Winooski, VT, USA). Growth was monitored for 19 h by changes in OD_600_. The MIC was defined as the lowest concentration of antimicrobial agent needed to completely inhibit the bacterial growth at 8 h (MIC_8h_). After the incubation period, 3 µL of the cultures used for the MIC assessment were droplet-plated in MH-agar for determination of the MBC. The plates were incubated for 24 h at 37 °C, and the MBC was defined as the lowest concentration where no colonies, indicative of cell growth, were observed.

### 2.7. Microscopic Analysis of Lipid Domain Formation

The Nile Red fluorescent lipid dye was used to visualize lipid domain formation of the *B. subtilis* cytoplasmic membranes upon BDTL100 (green fluorescently labeled form of BDTL049) treatment. The experiment was performed in MH media following previously described protocols [29,30]. Briefly, overnight cultures prepared as previously described were used to inoculate on the next day 10 mL day cultures (without antibiotic selection) with a 1:200 dilution. The cells were grown to exponential phase (OD_600_ 0.4–0.7), then diluted to an OD_600_ of 0.2 and exposed to BDTL100 at a final concentration of 4 µg mL^−1^, and further incubated for 20 min, shaking. The cells were stained with Nile Red (final concentration 1 µg mL^−1^) during the last 5 min of incubation prior to microscopy. For this, 2 µL samples were deposited onto microscope slides with agarose pads (1% UltraPure Agarose, Invitrogen) and air-dried at room temperature for 10 min. 

Fluorescence microscopy was performed using an Axio Observer.Z1/7 inverse microscope with Plan-Apochromat 100x/1.40 Oil DIC M27 objective and ZEN 2.3pro software (Carl Zeiss, Jena, Germany). Three channels were used with established parameters as follows. Bright-field contrast method: light source, TL LED, 3.00 Volt; exposure time, 4.6 s; depth of focus, 0.85 µm. Green fluorescence (eGFP) contrast method: light source, LED-Module 475 nm at 100% intensity; illumination wavelength, 450–488 nm (Ex: 488 nm/EM: 509 nm); exposure time, 800 ms; depth of focus, 0.79 µm. Red fluorescence (mCherry) contrast method: light source, LED-Module 567 nm at 100% intensity; illumination wavelength, 577–604 nm (Ex: 587 nm/EM: 610 nm); exposure time, 800 ms; depth of focus, 0.94 µm. Pictures were made with an AxioCam 702m left camera (Carl Zeiss, Jena, Germany). 

Microscopy pictures were analyzed using the tools implemented in the MicrobeJ (ImageJ 1.52 i) software [31,32].

### 2.8. Toxicity to Eukaryotic Cells

Cell viability was analyzed by MTT assay in PC3, MFC7, and HT29 cell lines. The cells were harvested into 24-well plates (1.0 × 10^4^ cells/well) and culture for 72 h. Afterward, they were treated with the compound BDTL049 in a range of concentrations, 0.1 to 4 μM, and cultured for 24 h at 37 °C. After the treatment, a 3-(4,5-dimethyl-2-thiazolyl)-2,5-dephenyl-2H-tetrazolium bromide (MTT) solution was added in the plates and incubated for 4 h. Finally, the cell culture was removed, and the formazan crystals were dissolved in DMSO. The optical density, proportional to the living cells of each plate, was measured using a microplate reader. Cell viability was calculated as a percentage of viable cells with respect to the vehicle-treated sample, which was assigned 100 % viability.

### 2.9. Statistics

Growth, luminescence, and MIC measurements were performed at least in biological duplicates and technical triplicates. From the values obtained for each time point, mean values and standard deviation (±) were calculated and plotted. For the microscopy analysis, pictures from independent biological clones were analyzed, and regions of interest (i.e., cells and maxima) were segmented by the algorithm (MicrobeJ) and manually corrected if necessary [31,32]. Analysis of BDTL049 toxicity towards eukaryotic cells was performed in duplicates for PC3 cell lines and in triplicates for MCF7 and HT29 cell lines.

## 3. Results

### 3.1. Chemical Synthesis of a Green-Fluorescently Labeled Form of BDTL049 and Validation of Its Functionality In Vivo

Our previous results indicated that the cytoplasmic membrane is the primary cellular target of the novel antimicrobial compound BDTL049, which depolarizes the membrane and leads to pore formation [18]. We next aimed at visualizing the interaction of the compound with the bacterial cell in vivo. For this, we created a green-fluorescently labeled form of BDTL049, termed BDTL100. Prior to performing microscopic studies, we investigated if the new compound retained similar activity against *B. subtilis* cells as the parental dendrimer. 

#### 3.1.1. Synthesis of BDTL100, the Green-Fluorescently Labeled Form of BDTL049

Carbosilane bow-tie dendrimer BDTL049 was synthesized in two simple steps (see Figure 1), as described previously [18]. BDTL049 contains 4 NHMe_2_^+^ Cl^-^ functional groups, and it has a molecular weight of 981.38 g mol^−1^. The compound is air and water stable, soluble in protic solvents such as water, methanol, or DMSO, and non-soluble in organic solvents. For the present work, a green-fluorescently labeled derivative (BDTL100) was also prepared following an adapted protocol for synthesis (see Experimental Section). BDTL100 contains 3 NHMe_2_^+^ Cl^−^ functional groups and 1 Fluorescein isothiocyanate (FITC) group, and it has a molecular weight of 1320.26 g mol^−1^. The compound is also soluble in protic solvents such as water or DMSO, but it is light-sensitive. The structure of the bow-tie dendrimer BDTL049 and its green-labeled form is shown in Figure 1 for comparison.

#### 3.1.2. Investigation of the Antimicrobial Activity of BDTL100

The antimicrobial properties of BDTL049 rely on its chemical composition and 3D structure. Prior to performing any microscopic studies, we performed assays with BDTL100 to validate that the addition of the FITC label with the concomitant elimination of one NHMe_2_^+^ Cl^-^ functional group did not impair the compound in functionality or activity. 

Firstly, we investigated the toxicity. For that, *B. subtilis* W168 exponentially growing cultures were exposed to increasing 2-fold dilutions of the compound (from 0.375 to 3 μg mL^−1^), and the effect on the bacterial growth was assessed by monitoring the changes in OD_600_ over time (Figure 2A). No effect was observed for the two lowest concentrations tested of BDTL100 as compared with growth in reference conditions. However, a concentration of 1.5 μg mL^−1^ impaired *B. subtilis* growth without completely inhibiting it, indicating most probably the presence of a heterogeneous affectation at the population level with different degrees of cellular damage present in the same culture. A concentration of 3 μg mL^−1^ completely impaired cell growth resulting in massive cell lysis. Importantly, these results mimic the toxicity results of the unlabeled parental compound BDTL049 (Figure 4Bi and [18]). 

Secondly, we investigated if BDTL100 retains the same mode of action as BDTL049. For that, we analyzed the response of the BDTL049-responsive envelope-stress inducible-promoters P*_liaI_* and P*_pspA_* to increasing concentrations of BDTL100 by monitoring their luminescence readout relative to the bacterial growth over time. After compound addition, both reporters showed comparable induction to the previously tested BDTL049 [18]. BDTL100 showed no induction of the whole-cell biosensors with the lowest concentrations tested of (0.375 and 0.75 μg mL^−1^). However, induction was observed when the damage-inducing threshold concentration of 1.5 μg mL^−1^ was reached (Figure 2B–E). Satisfactorily, the induction obtained with BDTL100 mimicked the induction pattern of the P*_liaI_* and P*_pspA_* whole-cell biosensors by BDTL049 regarding both the biosensor sensitivity (1.5 μg mL^−1^ of the inducing compound) and the induction intensity (10^5^ and 10^6^ RLU/OD_600_, for P*_pspA_* and P*_liaI_*, respectively) (Figure 2B–E and [18]).

These results indicate that BDTL100 retains the same antimicrobial activity against *B. subtilis* and induces equal cell envelope stress comparable to BDTL049. In line, demonstrating that the labeling does not modify the toxicity or potentially the mode of action of the parental compound. Supported by these observations, we next used BDLT100 for fluorescence-microscopy localization studies on a single-cell level.

### 3.2. BDTL100 Modifies Lipid Packing of the Cytoplasmic Membranes

Our previous microscopic results indicated that the activity of BDTL049 at the membrane is remarkably heterogeneous between different cells of the population [18]. To investigate the interaction between the synthetic antibiotic and the cell surface at the single-cell level, fluorescence microscopy was applied. For that, we used the green fluorescently labeled form BDTL100 (Figure 1), and the cytoplasmic membranes were labeled with the fluidity-sensitive membrane dye Nile red. Nile red is very soluble in organic solvents and rather insoluble in aqueous polar environments and has been described as a proxy for alterations in membrane fatty acid lipid packing in various studies [29,33,34]. Therefore, Nile red fluorescence is only observed within the bacterial membranes, while it is quenched in the surrounding media. 

*B. subtilis* W168 was treated for five or 20 min with 4 µg mL^−1^ of BDTL100. To visualize the effect of the antibiotic on membrane integrity, Nile red was added as a proxy, either simultaneously or five minutes prior to microscopy (in the case of 20 min BDTL100 treatment). 

Incubation for five minutes in total revealed cell damage indicated by Nile red patches at the poles and near the septa (Appendix A). These observations were profound for 20 min BDTL100 treatment, accompanied by the presence of membrane patches across the entire cell body and the appearance of dead cells (Figure 3A). A co-occurrence analysis of the labeled antibiotic and intensity profile of Nile red staining depicted a sharp overlap (Figure 3B,C). Quantification of this phenomenon (>200 cells) revealed that approximately 84% of Nile red maxima were covered by BDTL100 accumulation, whereas the remaining 16% of BDTL100 maxima did not correlate with Nile red intensification (Figure 3D). The non-colocalized fraction of BDTL100 and Nile red could hint towards a delay of the antibiotic mode of action at the sight of the membrane and Nile red accumulation. This is supported by the observed increase of antibiotic-caused Nile red patches comparing 5 and 20 min treatments, respectively. 

Overall, the fluorescence-labeled BDTL100 allows us to monitor and correlate antibiotic-induced formation of Nile red patches indicative of severe membrane damage due to altered lipid fatty acid packing. 

### 3.3. Mechanisms of Resistance against BDTL049

Mimicking the properties of many natural antibiotics, BTDL049 contains four net positive charges at its periphery (Figure 1), which is supposed to favor its interaction with the negatively charged bacterial cell envelope. Considering its ability to integrate within the cell membranes (Figure 3 microscopy), we hypothesized that the natural cell envelope stress response systems from *B. subtilis*, involved in response to other positively charged antibiotics also targeting the cell membrane (such as antimicrobial peptides), could also mediate the cell envelope response against the new synthetic antibiotic. Hence, we decided to investigate the sensitivity towards BDTL049 of a collection of *B. subtilis* mutants eliminated on cell envelope stress resistance determinants, probing their potency to counteract the antimicrobial activity of the novel synthetic compound. Besides mutant strains eliminated in the PspA and LiaIH systems, mutant strains eliminated in the UPP phosphatase BcrC, the DltABCD operon, and the MprF system (Table 1) were also tested for their sensitivity against BDTL049. PspA and LiaIH are phage shock proteins involved in the bacterial envelope stress response to numerous antibiotics [35,36]. BcrC catalyzes the dephosphorylation of the lipid carrier undecaprenyl pyrophosphate (UPP) to undecaprenyl phosphate (UP), ensuring the progression of the Lipid II cycle for cell wall biosynthesis [37] and maintaining cell wall homeostasis [22]. The DltABCD system catalyzes the D-alanylation of teichoic acids on the cell wall [26,38,39], and the MprF protein catalyzes the lysinylation of membrane phospholipids [40]. The activity of the DltABCD and MprF systems decreases the net negative charge of the cell wall and the cell membrane, respectively. This provides bacterial resistance to antibiotics such as cationic antimicrobial peptides by preventing them from reaching their molecular targets at the surface of the cytoplasmic membrane (reviewed in [41]). 

Fresh-day cultures of the strains under study were incubated with increasing two-fold concentrations of BDTL049, and the MIC_8h_ and the MBC after the 19h incubation period in the presence of the compound were calculated and compared with the values for the parental strain W168 [18]. The results showed that from the five single mutants tested, *∆liaIH*, *∆pspA*, *∆bcrC*, *∆dltABCD,* and *∆mprF*, only the mutant in the Dlt system was slightly more susceptible to BDTL049 than the wild-type (MIC_8h_ between 2–4 µg mL^−1^ for *∆dltABCD* and 4 µg mL^−1^ for W168) (Figure 4A). Interestingly, previous reports investigating the response of *B. subtilis* to the antimicrobial peptide bacitracin showed that the involvement of some of the resistance systems mediating the different layers of the cell envelope stress response was only evident upon the removal of several systems simultaneously within a single strain [22]. Hence, we next tested the sensitivity against BDTL049 of double and triple mutants in the systems under study, aiming at identifying potential redundant or multi-layered resistance networks against BDLT049 action. The results showed that only the strains *∆liaIH∆dltABCD*, *∆dltABCD∆mprF*, *∆liaIH∆dltABCD∆mprF*, and *∆liaIH∆pspA-ydjGHI∆dltABCD* depicted a slight increase in sensitivity towards BDTL049 and in all cases the sensitivity compared to the single *∆dltABCD* mutant strain (Figure 4A; MIC_8h_ values between 2–4 µg mL^−1^).

Regarding the MBC values, all the mutant strains had an MBC of 4 µg mL^−1^, whereas the MBC for W168 was between 4–8 µg mL^−1^.

Being BDTL049 a positively charged molecule, these MIC_8h_ results suggest that the novel compound is impaired in reaching its target/docking molecule(s) within the cell membrane when the Dlt is functional. Since these differences in MIC_8h_ and MBC values between the sensitive strains and the parental strain were only indicative rather than conclusive (Figure 4A), we verified the results by testing the sensitivity of exponentially growing cultures of these strains against increasing concentrations (0.375 to 3 µg mL^−1^) of BDTL049 (Figure 4B). All the strains were lysed at the highest concentration tested. When exposed to a BDTL049 concentration of 1.5 µg mL^−1^, the growth of exponentially growing cultures of the *∆dltABCD* strain was more impaired than the growth of W168 (Figure 4B(*i*,*ii*), in agreement with the slightly different MIC values for these two strains. However, removal of other resistance determinants over a *∆dltABCD* background (strains *∆liaIH∆dltABCD*, *∆dltABCD∆mprF*, *∆liaIH∆dltABCD∆mprF*, and *∆liaIH∆pspA-ydjGHI∆dltABCD*) did not result in an increased sensitivity of these strains against BDTL049 relative to the single *∆dltABCD* mutant (graphs *ii* to *vi* in Figure 4B). The lack of additive effects in the double and triple mutants indicates that from all the general, unspecific-resistance determinants tested here, only the activity of the Dlt system, which reduces the cell wall negative surface charge [38], partially hinders the positively-charged BDTL049 compound from reaching its cellular target(s). 

### 3.4. Cytotoxicity of BDTL049 in Human Cells

The potential application as a new antimicrobial agent of any new rationally designed compound, such as BDTL049, will mostly depend on its biocompatibility and lack of toxicity against human cells. Therefore, we investigated the cytotoxicity of BDTL049 by performing MTT assays on different cell lines (PC3, MCF7, and HT29 cell lines). Surprisingly, all the eukaryotic cell lines used showed toxicity in the range 1–1,5 μM (Appendix A), very similar to that observed for *B. subtilis.*

## 4. Discussion

Rationally designed synthetic antibiotics comprise but one strategy to combat the ever-increasing threat in modern medicine when it comes to treating multi-drug resistant bacterial infections [1,42]. In this study, we present an updated view of a previously introduced dendritic compound: BDTL049, that exhibits strong antibacterial properties [18]. The bow-tie dendrimer kills off *B. subtilis* by dissipating membrane potential and causing severe membrane perturbations [18]. In this study, we further investigated the mode of action of BDTL049. To get a better understanding of how the membrane is damaged by the compound, a fluorescent-labeled version: BDTL100, was synthesized (Figure 1). We confirmed BDTL100 antibacterial capacity by exposing *B. subtills* to the same concentrations as BDTL049 and observing an adequate response of the cell envelope stress response systems *pspA* and *liaH* (Figure 2). As discussed previously, induction of these systems can be considered to align with general envelope stress, and most likely, *B. subtilis* aims to counteract the damage caused rather than tackle the compound directly [18,43]. Having a fluorescent-labeled compound at hand, we could now further elaborate on the mode of action of BDTL049. Nile red-stained bacteria exposed to BDTL100 depicted patches of Nile red co-localized with compound fluorescence (Figure 3). Nile red is well described to clump upon disturbance of membrane fluidity, including alterations of local fatty acid lipid packing [44]. These lipid domain formations have been previously described as an antibacterial strategy of natural antimicrobial peptides produced by *B. subtills* as well as for the last-resort antibiotic daptomycin [29,33]. Consequently, the bacterium faces upon BDTL049 (or BDTL100) exposure, dissipation of membrane potential, pore formation, as well as severe alteration of membrane lipid composition, highlighting the diverse antimicrobial portfolio of the synthetic compound. This massive membrane-centered mode of action leaves the bacteria rather defenseless, as the prime resistance determinants in our mutational study showed no increase in sensitivity (Figure 4). The only exception is the Dlt system, with a modest contribution. This physiologically important machinery centers crucially in Gram-positive bacteria modulating the cell wall as it encodes the D-alanine incorporation system [38]. As the most prominent phenotype, deletion of Dlt impairs the ability of the cell to decrease the cell surface net negative charge. Thus, an apparent hypothesis of slight sensitivity observed in the Dlt deletion strain towards the positively dendritic BDTL049 lies in the decrease in repellence of charges between the bacterium and compound, which could also make the cell surface more permeable to BDTL049 [41]. Alternatively, challenging L-forms, i.e., cell-wall deficient bacteria with BDTL049, could provide further insights into the exact interactions between this specific section of the cell envelope and the antimicrobial. It has been demonstrated that cell envelope targeting antibiotics require an intact cell wall for full antimicrobial effectiveness, whereas there is no evidence so far that for the synthetic antibiotic, this must also hold true [45]. Having now comprehensively described the bacterial side of the BDTL049 exposure, a natural next step comprised the exposure to eukaryotic cell lines. Cytotoxicity resembles the first step of probing the applicability of BDTL049 with clinically relevant purposes. However, BDTL049 was revealed to have toxicity very similar to that shown for bacteria (Appendix A). Although this can be seen as a drawback, the proof of concept here achieved, concerning its very good antibacterial activity, makes it necessary to improve the synthetic design for affording new related compounds to reduce toxicity while maintaining the effectiveness towards bacteria. In this sense, future works are in progress to prepare analogue bow-tie dendrimers modifying the nature of the ammonium groups [-NHMe_2_]^+^ for more lipophilic units like [-NMe_3_]^+^ or more general formulations of type [-NMe_2_R]^+^. This behavior has been observed elsewhere in ammonium-terminated spherical carbosilane dendrimers [12]. Furthermore, testing of the antibacterial capacity of BDTL049 towards the consortia of bacteria or cells within the multicellular environment, such as biofilms on surfaces, should also be considered in future studies. 

## 5. Conclusions

Synthetic, rationally designed molecules to combat the antibacterial crisis in modern medicine represent a promising path. Here, we elaborated on a previously described dendritic compound: BDTL049. Although we could highlight and further characterize the effectiveness against the Gram-positive model organism *B. subtilis,* a drawback was observed when tested against human cell line cells. The cytotoxicity of the compound reported here, however, should not serve as an obstacle but rather trigger further rounds of developments based on cationic carbosilane derivatives due to their tremendous effectiveness in antimicrobial activity. 

## Figures and Tables

**Figure 1 pharmaceutics-15-00297-f001:**
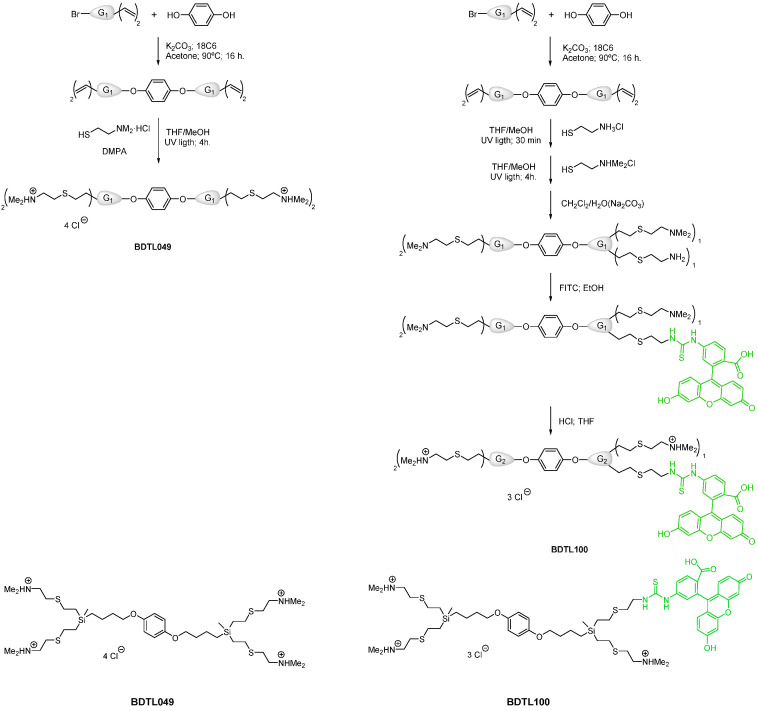
Synthesis and structure of the cationic carbosilane derivatives used in this study. Upper-left, the synthetic procedure for the formation of compound BDTL049 as previously described [18]. Upper-right, the synthetic procedure for the formation of the green-fluorescently labeled form BDTL100. The final chemical structures of the original BDTL049 and derivative compound BDTL100 are shown in the lower panel.

**Figure 2 pharmaceutics-15-00297-f002:**
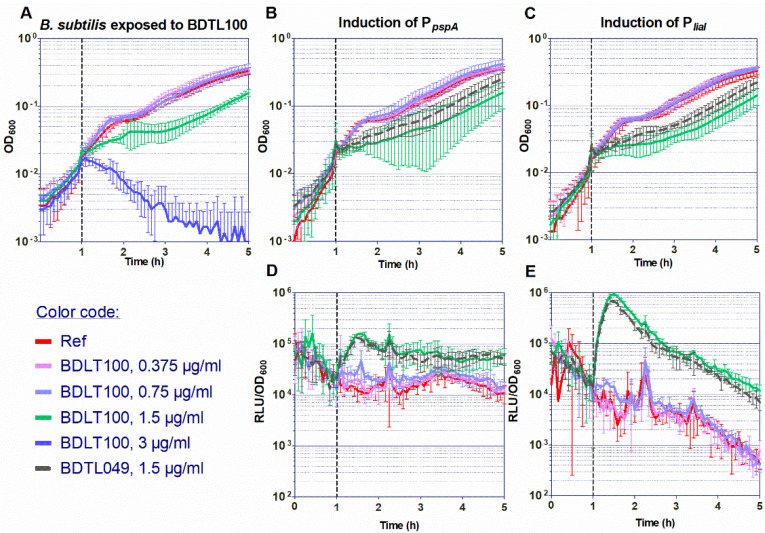
Sensitivity of *B. subtilis* cells toward BDTL100, and comparison of the induction of cell-envelope stress-reporter strains by BDTL100 and BDTL049. (**A**) The effect of the addition of increasing concentrations of BDTL100 on the growth of exponentially growing wild-type cells of *B. subtilis* W168 was determined by monitoring OD_600_ over time. The time of antibiotic addition is indicated by a vertical black dashed line. (**B**–**E**) Induction of the P_pspA_ (**B**,**D**) and P_liaI_ (**C**,**E**) promoters by BDTL100 in liquid media. For comparison, the induction of the P_pspA_ and P_liaI_ promoters by BDTL049 at a concentration of 1.5 µg mL^−1^ previously reported [18] is also presented (dashed line). The effect of antibiotic exposure on growth is indicated as OD_600_ (**B**,**C**), and promoter induction as relative luminescence units by OD_600_ (D,E). The time of antibiotic addition is indicated by vertical black dashed lines as in (**A**). The compounds’ concentrations used are indicated below in graph (**A**). The results presented in (**B**–**E**) correspond to strains TMB2299 (P_pspA_) (**B**,**D**) and TMB3822 (P_liaI_) (**C**,**E**). All experiments were performed at least in triplicate. Means and SDs are depicted.

**Figure 3 pharmaceutics-15-00297-f003:**
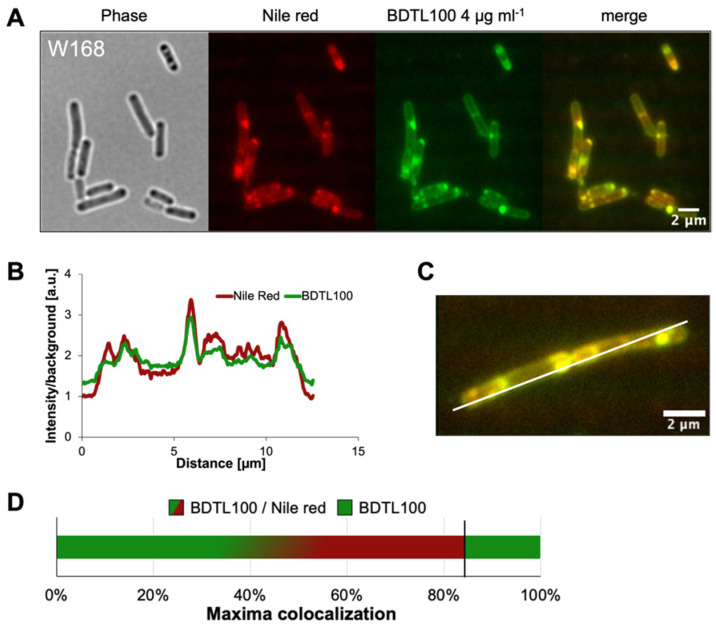
BDTL100 disturbs membrane homeostasis by inducing lipid domain formation in vivo. (**A**) *B. subtilis* W168 cultures were treated for 20 min (for 5 min treatment see (Appendix A) with 4 µg mL^−1^ BDTL100, the green fluorescently labeled version of BDTL049 (Figure 1). The panel shows phase contrast, Nile red, and BDTL100 fluorescence microscopy channels and the merge. (**B**) Fluorescent intensity profile of both Nile red and BDTL100 channels across the white line in (**C**). (**D**) Quantification of BDTL100 and Nile red fluorescence maxima. Co-localization was observed for more than 84% percent, whereas as the remaining 16% of BDTL100 accumulations lacked a corresponding Nile red intensification. Scale bars are set to 2 µm.

**Figure 4 pharmaceutics-15-00297-f004:**
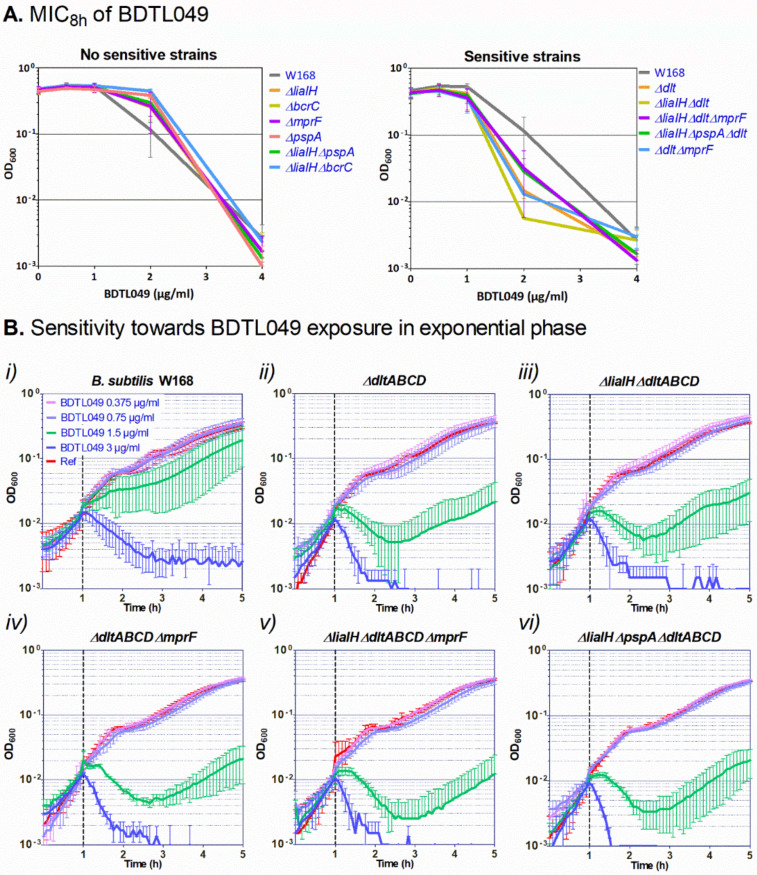
Sensitivity assays of *B. subtilis* W168 and derivative mutant strains exposed to the synthetic antibiotic BDTL049. (**A**) MIC after 8 h of growth in the presence of BDTL049. The cells were inoculated to an OD_600_ of 0.05 in MH with 2-fold serial dilutions of BDTL049, and the cultures were incubated in aerobic conditions at 37 °C. The values of OD_600_ after 8 h of incubation in the presence of the compound are plotted as a function of the antibiotic concentration to determine the MIC_8h_. Results present the mean and standard deviation of three replicas. (**B**) The effect of the addition of increasing concentrations of BDTL049 on the growth of exponentially growing cells was determined by monitoring OD_600_ over time. The time of compound addition is indicated with a vertical black dashed line. The compound concentrations are indicated in graph *(i*). The results presented correspond to strains: *(i)*, *B. subtilis* W168; *(ii)*, *∆dltABCD* (TMB5498); *(iii)*, *∆liaIH∆dltABCD* (TMB5499); *(iv)*, *∆dltABCD∆mprF* (TMB5512); *(v)*, *∆liaIH∆dltABCD∆mprF* (TMB5513); *(vi)*, *∆liaIH∆pspA-ydjGHI∆dltABCD* (TMB5500). The experiments were performed at least in triplicate. Means and standard deviations are depicted.

**Table 1 pharmaceutics-15-00297-t001:** Bacterial strains and vectors used in this study.

Strain or Vector	Description ^1^	Source/ Reference
*B. subtilis* reference strain		
W168	Wild type; trpC2	Lab collection
*B. subtilis* mutant strains		
TMB0297	W168 *bcrC::tet^r^*	[20]
TMB1151	W168 *∆liaIH*	[21]
TMB1706	W168 *pspA-ydjGHI::mls^r^*	Lab collection
TMB1718	W168 *∆liaIH//pspA-ydjGHI::mls^r^*	Lab collection
TMB2128	W168 *ΔliaIH//bcrC::tet^r^*	[22]
TMB5498	W168 *dltABCD::kan^r^*	This study
TMB5499	W168 *∆liaIH//dltABCD::kan^r^*	This study
TMB5500	W168 *∆liaIH//pspA-ydjGHI::mls^r^ //dltABCD::kan^r^*	This study
TMB5511	W168 *mprF::mls^r^*	This study
TMB5512	W168 *dltABCD::kan^r^//mprF::mls^r^*	This study
TMB5513	W168 *∆liaIH//dltABCD::kan^r^//mprF::mls^r^*	This study
*B. subtilis* Reporter strains		
TMB2299	W168 *sacA:: cm^r^* pASp3C*lux01* (P*_pspA_*-*lux*)	Lab collection
TMB3822	W168 *sacA:: cm^r^* pBS3C*lux*-P*_liaI_*	[23]
Vectors		
pDG647	pSB119, *erm^r^*	[24]
pDG783	pSB118, *kan^r^*	[24]

^1^ *cm^r^*, chloramphenicol resistance; *kan^r^*, kanamycin resistance; *mls^r^*, macrolide-lincosamide-streptogramine resistance, *tet^r^*, tetracyclin resistance, *erm^r^*, erythromycin resistance.

**Table 2 pharmaceutics-15-00297-t002:** Oligonucleotides used in this study for gene knockout via Long-Flanking-Homology.

Nº in Collection	Name	Description (Sequence) ^1^	Use
TM0137	kan-fwd	CAGCGAACCATTTGAGGTGATAGG	Kanamycin cassette
TM0138	kan-rev	CGATACAAATTCCTCGTAGGCGCTCGG	
TM0056	kan-check-fwd	CATCCGCAACTGTCCATACTCTG	Check allelic replacement
TM0147	kan-check-rev	CTGCCTCCTCATCCTCTTCATCC	
TM0139	mls-fwd	CAGCGAACCATTTGAGGTGATAGGGATCCTTTAACTCTGGCAACCCTC	*mls* cassette
TM0140	mls-rev	CGATACAAATTCCTCGTAGGCGCTCGGGCCGACTGCGCAAAAGACATAATCG	
TM0148	mls-check-rev	GTTTTGGTCGTAGAGCACACGG	Check allelic replacement
TM0057	mls-check-fwd	CCTTAAAACATGCAGGAATTGACG	
TM2895	P*_liaI_* fwd EcoRI NotI XbaI	GATCGAATTCGCGGCCGCTTCTAGAGATTGGCCAAAGCAGAAAGGTCC	Construction of pBS3Clux-P_liaI_
TM2896	P*_liaI_* rev SpeI	GATCACTAGTATCGTTTTCCTTGTCTTCATCTTATAC	Construction of pBS3Clux-P_liaI_
TM3268	P*_pspA_* EcoRI fwd	ttataggaattccgcggccgcttctagagTCCGGTGACATCAATTGACTC	Construction of pASp3C*lux01* (P*_pspA_*-*lux*)
TM3269	P*_pspA_* SpeI rev	ctataaactagtAAAGCTAATTCGGTAACCCTTG	Construction of pASp3C*lux01* (P*_pspA_*-*lux*)
TM5175	up-fw-dltA-LFH	CGTTTTAGGCTTCATTCCGTG	*dltABCD* up fragment
TM5176	up-rv-dltA-LFH	CCTATCACCTCAAATGGTTCGCTGGTTTCCGCATGTGTTTGAATAG	
TM6088	down-fw-dltD-LFH	CGAGCGCCTACGAGGAATTTGTATCGCTGGGTGTATGTCGATAAAGC	*dltABCD* down fragment
TM6089	down-rv-dltD-LFH	CATGGTCAATCTCCCTGCTG	
TM6111	LFH-mprF-up-fwd	AGTCCGAACAGGCAAACC	*mprF* up fragment
TM6112	LFH-mprF-up-rev	CCTATCACCTCAAATGGTTCGCTGAGGAAAAACGATTTTTAATATTGATAAAGC	
TM6113	LFH-mprF-do-fwd	CGAGCGCCTACGAGGAATTTGTATCGACACGTCTGATTGGCAAAAGC	*mprF* down fragment
TM6114	LFH-mprF-do-rev	AGGGATTGACACTCTTAACACTG	

^1^ Sequences are given in the 5′→ 3’direction. The sequences underlined are inverse and complementary to the 5′ (up-reverse), and 3′ (do-forward) ends of the *kan* cassette and *mls* cassettes, respectively.

## Data Availability

Not applicable.

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
