# Peer review of "The Novel Synthetic Antibiotic BDTL049 Based on a Dendritic System Induces Lipid Domain Formation while Escaping the Cell Envelope Stress Resistance Determinants"

_pharmaceutics, 2023, doi:10.3390/pharmaceutics15010297_

Round 1

Reviewer 1 Report

This manuscript describes a modern approach for testing the suitability of a previously chemically synthesized product as a novel antibiotic. The authors used many different approaches, constructing mutants, fluorescently labeled chemicals, and transcriptional fusion to study the effect of this novel chemical. Generally, I am supporting the manuscript for publication. However, the authors have to add to the methods a procedure of construction luxCDBAE transcriptional fusions, check English throughout the text, and put the Latin names of bacteria into italic.  

Author Response

We thank Reviewer 1 for highlighting and proposing improvements to the Manuscript. We have revised it accordingly. In detail: English changes were applied throughout the text to avoid long and confusing sentences to allow the reader a better flow. Individual passages were adjusted. The construction of the lux-reporters is now part of the M&M section (2.4) with primers used in Table 2. We have checked the Mns for bacterial species names and made them italics. 

Reviewer 2 Report

On account of the manuscript PHARMACEUTICS-2131715, entitled “The Novel Synthetic Antibiotic BDTL049 Based on a Dendritic System induces lipid domain formation while escaping the cell envelope stress resistance determinants” by Philipp F. Popp et al., the authors synthesized a fluorescent-labeled version of antibiotic BDTL049 and then evaluated its bacterial sensitivity with cytotoxicity. The topic is important to develop the potent antimicrobial molecules, and to conduct human health management for antimicrobial resistant bacteria as well. The manuscript was well written and designed, and the authors got interesting results. After careful consideration, I made a decision that the manuscript is acceptable for publication in its present form.

Special remarks:

‧ The present manuscript evaluated the characteristics and effect of antibiotic BDTL049 for treatment of antimicrobial resistant bacteria.

‧ The authors designed novel antibiotic BDTL049 for the knowledge-based strategy which seeks the development of increasingly potent antimicrobial molecules, is considered to new view point and interesting.

‧ The present manuscript provided useful prospects to better understandings for the potent antimicrobial molecules and human health risk assessment.

‧ The interpretation of the evidence and arguments presented and conclusions are sufficient.

‧ The references cited relevant and up to date.

‧ The tables and/or figures are useful, necessary, and good quality.

Author Response

We thank Reviewer 2 for his/her positive evaluation of our Manuscript. 

Round 2

Reviewer 1 Report

Please, correct the writing of the genes under paragraph 2.3 into italic. 

Author Response

Thank you for pointing this out, I have overlooked them.